# Synthesis of SAPO-34 Nanoplates with High Si/Al Ratio and Improved Acid Site Density

**DOI:** 10.3390/nano11123198

**Published:** 2021-11-25

**Authors:** Syed Fakhar Alam, Min-Zy Kim, Aafaq ur Rehman, Devipriyanka Arepalli, Pankaj Sharma, Churl Hee Cho

**Affiliations:** 1Reaction and Separation Nanomaterials Laboratory, Graduate School of Energy Science and Technology, Chungnam National University, 99 Daehak-ro, Yuseong-gu, Daejeon 34134, Korea; fakhar9689@o.cnu.ac.kr (S.F.A.); mzkim@cnu.ac.kr (M.-Z.K.); aafaqktk@cnu.ac.kr (A.u.R.); devipriya0508@o.cnu.ac.kr (D.A.); 2Department of Chemistry, Cardiff University, Park Place, Cardiff CF10 3AT, UK; SharmaP14@cardiff.ac.uk

**Keywords:** SAPO-34, microwave hydrothermal synthesis, Si/Al ratio, nanoplates, density of acid sites

## Abstract

Two-dimensional SAPO-34 molecular sieves were synthesized by microwave hydrothermal process. The concentrations of structure directing agent (SDA), phosphoric acid, and silicon in the gel solution were varied and their effect on phase, shape, and composition of synthesized particles was studied. The synthesized particles were characterized by various techniques using SEM, XRD, BET, EDX, and NH_3_-TPD. Various morphologies of particles including isotropic, hyper rectangle, and nanoplates were obtained. It was found that the Si/Al ratio of the SAPO-34 particles was in a direct relationship with the density of acid sites. Moreover, the gel composition and preparation affected the chemistry of the synthesized particles. The slow addition of phosphoric acid improved the homogeneity of synthesis gel and resulted in SAPO-34 nanoplates with high density of acid sites, 3.482 mmol/g. The SAPO-34 nanoplates are expected to serve as a high performance catalyst due to the low mass transfer resistance and the high density of active sites.

## 1. Introduction

Aluminophosphates (AlPO) are a special class of zeolites based on the tetrahedra of AlO_4_ and PO_4_. If phosphorus is replaced by silicon in the structure, silicoaluminophosphate (SAPO) with a negative charge and Brønsted activity is obtained [1,2]. These molecular sieves have a molecular formula of Si*_x_*Al*_y_*P*_z_*O_2_ where *x* = 0.01–0.98, *y* = 0.01–0.60, *z* = 0.01–0.52, and *x* + *y* + *z* = 1 [3]. Many different SAPOs having a wide variety of structures, including AEI, AFI, and CHA have been reported by researchers [4]. SAPO-34 has a CHA-type structure with eight ring channels and a uniform pore size of 3.8 × 3.8 Ǻ [5,6,7]. Researchers have reported it in a variety of applications, including catalysts [8,9], heat and thermo-chemical storage materials [10,11], and synthesized as gas separation membranes for natural gas sweetening [6,12,13]_,_ H_2_ enrichment [14,15], and noble gases collection [16,17].

Particulate SAPO-34 zeolites are widely reported as potential catalysts for the methanol to olefin (MTO) process due to their higher C-2 and C-3 selective nature compared with the MFI zeolite [18]. However, due to the smaller pore channel of SAPO-34, it has higher mass transfer resistance, which increases the coke formation and reduces the catalyst life [19,20]. As a result, researchers have tried to reduce the size of isotropic SAPO-34 catalysts to overcome this issue. Hajfarajollah et al. elucidated that the aggregated SAPO-34 particles had higher catalytic performance and lifetime for MTO reaction compared with coarse isotropic particles [21]. Similarly, Salmasi et al. used a dual-template to reduce the particle size of SAPO-34 cubes and reported a longer catalytic lifetime and improved activity [18]. Another way to improve catalytic activity is to alter the morphology of SAPO-34 particles. Sun et al. synthesized the hierarchical particles by the seed-assisted synthesis method and reported better performance [22]. Wang et al. and Haifarajollah et al. synthesized aggregated micro particles and were able to achieve better catalytic performance [21,23]. However, due to the relatively large particle size, coking remained a major issue, and the catalytic lifetime was still limited.

Numerous researchers tried to synthesize the SAPO-34 nanoplates in order to reduce the mass transfer resistance. Lin et al. presented the effect of silica precursor on morphology for SAPO-34 particles and explained that colloidal silica is a preferred precursor to obtain sheet-like morphology with minimal aggregation [24]. Muñoz et al. used microwave-assisted synthesis to synthesize high purity sheet-like SAPO-34 particles [25]. Hong et al. presented the effect of time on the development of plate-like SAPO-34 crystal [26]. Di et al. reported sheet-like SAPO-34 particles using seed-assisted synthesis [27]. Yang et al. were able to synthesize SAPO-34 plate like particles with an enhanced catalytic lifetime [8]. The Si/Al ratio of the SAPO-34 particles in these reports is lower compared with the isotropic cubic particles and hence is the catalytic activity. Therefore, the synthesis of plate like SAPO-34 particles with a higher Si/Al ratio is necessary to obtain high catalytic activity and long lifetime simultaneously.

The Si incorporation and Si/Al ratio in the AlPO structure determines the density of acid sites in SAPO-34. The amount of acid sites has a direct relationship with the catalytic performance [9,28,29,30]. Three different mechanisms can explain the Si substitution to generate acid sites in the SAPO framework. These mechanisms can be expressed as the following equations:[P^5+^] + Si^4+^ + H^+^ → [Si^4+^ + H^+^] + P^5^ → (SM-2)
[Al^3+^ + P^5+^] + 2Si^4+^ → [2Si^4+^] + P^5+^ + Al^3+^ → (SM-3)
[Al^3+^ + 4P^5+^] + 5Si^4+^ + 3H^+^ → [5Si^4+^ + 3H^+^] + Al^3+^ + 4P^5+^ → (SM-2 + SM-3)
[] represents the part of crystal structure.

The SM-2 mechanism is a single P replacement with a single Si to generate a proton in the framework and results in a weak Brønsted acid site as a result of single Si incorporation, which is surrounded by 4 Al atoms (Si(4Al)). A molar balance for the SM-2 substitution will occur for Si, P, and Al in the structure, *n*Si + *n*P = *n*Al with Si/Al ratio maintained at 0.25. In the case of the SM-3 mechanism, both P and Al are replaced with two Si. The Si generated in the SM-3 substitution results in the Si(3Al), Si(2Al), and Si(1Al) sites. This will result in a molar imbalance, *n*Si + *n*P > *n*Al. The third mode is a combination of both the SM-2 and SM-3 mechanisms. The 3SM-2 + SM-3 mode of substitution leads to combinations of various acid sites. The acid strength of these Si substitutions is in the following order:Si(4Al) < Si(3Al) <Si(2Al) < Si(1Al)

In the past, various researchers have studied the formation of the Brønsted acid sites in the SAPO-34 structure. Sastre et al. used computational models to explain the generation of the Brønsted acid sites by correlating them with the presence of silicon islands [31]. Tan et al. reported the crystallization and incorporation mechanism for SAPO-34 and experimentally observed a relationship between the acidity and the structure [32]. Suzuki et al. studied the proton formation in the SAPO-34 and found that the acid strength of the two prominent Brønsted acidic OH (surface and within the structure) was equal in the SAPO structure [33]. These studies highlight the importance of the effective acid sites in the SAPO-34 structure for catalytic activity. Moreover, these mechanisms show that a dominant P substitution is required to generate a large amount of acid sites in the structure, which is only possible with the well-dispersed incorporation of Si in the SAPO-34 structure. A concentrated substitution will increase Al substitution and also lead to the formation of large Si islands. Even with a higher Si/Al ratio in these particles, these Si islands will be catalytically inactive since there are no Brønsted acid sites, significantly reducing the performance and lifetime [30,33]. Therefore, Si substitution without large islands is necessary to avoid the compromise on catalyst activity.

In the present study, we aimed to synthesize the plate-like SAPO-34 particles with a high Si/Al ratio with the limited formation of Si islands. This was made possible by controlled mixing of precursors, which allows homogeneous gel formation. A quantitative analysis of the molecular tetrahedral was performed by using FTIR and the density of acid site, which is directly related to the catalytic activity studied.

## 2. Materials and Methods

### 2.1. Synthesis of SAPO-34 Particles

Commercially available precursors were used without further purification for the synthesis of SAPO-34 particles. Tetraethylammonium hydroxide (40 wt.% aqueous solutions, Sigma-Aldrich, St. Louis, MO, USA) was used as the template and aluminum isopropoxide (99% Al (O-i-Pr)_3_, Sigma-Aldrich, St. Louis, MO, USA), phosphoric acid (85 wt.% H_3_PO_4_, Junsei Chemical Co. Ltd., Tokyo, Japan), and Ludox colloidal Silica (40 wt.% SiO_2_ aqueous suspension, Sigma Aldrich, St. Louis, MO, USA) were used as aluminum, phosphorus, and silicon precursors, respectively.

Hydrothermal synthesis solutions were prepared directly in the glass vessel of a microwave hydrothermal reactor (Discover SP-Microwave Synthesizer, CEM Corporation, NC, USA). Initially, tetraethylammonium hydroxide and DI water were mixed and left to stir for 10 min. Aluminum iso-propoxide was then added and the mixture was stirred for 2 h at 323 K until the solid was completely dissolved to form a homogeneous solution. The solution was then brought back to room temperature. Colloidal silica was then added gradually into the homogeneous solution, and the solution was again left for another 2 h. Finally, phosphoric acid was dropwise added to the stirring mixture at a controlled rate, which gave a milky solution. The vessel was covered with the cap and the gel solution was left to age for 3 days at room temperature. All the samples were synthesized using a 22 g gel solution. After aging, the glass vessel containing the synthetic gel solution was directly transferred to the microwave reactor (Discover SP200, CEM Corporation, NC, USA). The temperature and time of the synthesis were adjusted, and the synthesis was carried out without stirring at a pressure of 1.72 × 10^6^ Pa and power of 150 W.

After the microwave hydrothermal synthesis, the as-synthesized product was separated and washed using a centrifuge machine (Supra 22k-Hanil Biomed high-speed centrifuge, Hanil Scientific Inc., Gwangju, Korea) at 5000 rpm in 25 cc polypropylene (PP) vessels, and particles were collected from the bottom. The process was repeated by dispersing the collected particles in DI water until there was no further change in the pH upon dispersion. The washed SAPO-34 particles were dried overnight at 373 K in the oven.

The dry particles were then calcined in a box furnace (51542-HR, Lindberg, Lindberg/MPH, MI, USA) at 823 K with a heating rate of 1 K/min for 5 h. Oxygen was maintained in the furnace at a flow rate of 25 cm^3^/min. After calcination, the furnace was allowed to cool back to room temperature with the flowing oxygen before the removal of the samples.

### 2.2. Characterization of SAPO-34 Particles

Various characterization techniques were used to analyze the as-synthesized particles after calcination.

X-ray diffraction (XRD) patterns were retrieved using a Philip X’pert X-ray diffractometer (Malvern Pananalytical Ltd, Malvern, UK) with Cu Kα radiation (λ = 0.15418 nm) at 40 mA and 40 kV with a scanning rate of 4° min^−1^. 

The morphology of the synthesized SAPO-34 particles was observed through a Hitachi S-4800 Cold Type Field Emission Scanning Electron Microscope (Hitachi, Tokyo, Japan) with an accelerating voltage of 10 kV and a current of 10 mA. 

The particle size was analyzed using differential light scattering (DLS) using the Microtrac Nanotrac Wave II analyzer (Microtract MRB, PA, USA). The SAPO-34 particles were dispersed in water and the suspension was sonicated for almost 15 min before it was used for the analysis. 

Energy disperse X-ray spectroscopy (EDX) by using a JSM-7000F field emission scanning electron microscope (JEOL Ltd., Tokyo, Japan) was used to analyze the composition. Each sample was tested at 5 points to gain reliable data, and the average was used for the calculation of molar Si/Al and P/Al ratios, and Si incorporation. The Si incorporation was calculated using the following equation:$$Si\;Incorporation=\frac{{\left[\frac{Si}{Si+Al+P}\right]}_{product}}{{\left[\frac{Si}{Si+Al+P}\right]}_{gel}}$$

Temperature programmed desorption (TPD) was performed using the BELCAT-M (BEL Japan, Inc., Osaka, Japan) catalyst analyzer using NH_3_ gas. The samples were pretreated at 773 K in a He atmosphere with a constant flow rate of 80 ccm. The samples were analyzed for NH_3_-TPD for a temperature range of 373–873 K with a constant heating rate of 10 K/min. 

The surface area and gas adsorption of the synthesized SAPO-34 particles were tested using the Microtrac BELSORP-max (Microtrac MRB, Tokyo, Japan) using nitrogen as the adsorption gas. The samples were pretreated at 573 K under vacuum for 24 h before the analysis.

## 3. Results and Discussion

The prime objective of this study is to synthesize SAPO-34 nanoplates with a high Si/Al ratio and improved Si incorporation, which is directly related to the density of acid sites, in other words, the catalytic activity. To reach the goal, we varied the concentrations of the template and phosphoric acid, the time for phosphoric acid addition for preparing the gel solution, and the hydrothermal time. The gel composition was 1Al_2_O_3_:*x*TEAOH:*y*P_2_O_5_:*z*SiO_2_:100H_2_O where *x* = 0.0–4.0, *y* = 1.0–2.0, and *z* = 0.4–0.6. The synthesis conditions for 17 samples synthesized in the present study are listed in Table 1. The crystalline phase, morphology, and Si/Al ratio of the synthesized particles are summarized in Table 2.

### 3.1. Effect of Template Concentration (x) on the Formation of SAPO-34 Particles

A total of 8 samples (S1–S8) were synthesized from the gel solutions with a composition of 1Al_2_O_3_:*x*TEAOH:2P_2_O_5_:0.6SiO_2_:100H_2_O where *x* was 0.0 to 4.0. XRD, SEM, and EDX results are shown in Figure 1, Figure 2 and Figure 3, respectively, and the DLS particle size distributions of these particles are presented in Appendix A. With no template (*x* = 0, S1), some silica and SAPO-5 peaks were observed in an amorphous phase in the XRD pattern. There was no visible crystal in the SEM image. As the template was introduced (*x* = 1.0, S2), sharp peaks of SAPO-5 at 2θ values of 7.9°, 20°, 23°, and 26° were observed [34]. In the SEM image, large aggregates of SAPO-5 with the typical hexagonal morphology were seen [35]. The average particle size of these aggregates was around 3.91 µm as shown by the DLS data (Appendix A). The low amount of template in the synthesis gel resulted in low Si incorporation in the AlPO structure, inducing the dominant growth of the SAPO-5 phase with a low Si/Al ratio [36].

Further template addition, i.e., *x* = 1.5–4.0 (S3–S8), resulted in the formation of a pure SAPO-34 phase. The XRD pattern revealed major characteristic peaks of SAPO-34 at 2*θ* values of 9.5°, 12.8°, 14.1°, 16.0°, 18.0°, 20.5°, 25.8°, 30.5°, and 34.4° [37,38]. The S3 formed with *x* = 1.5 shows perfect cubic morphology in the synthesized SAPO-34 particles. Both SEM (Figure 2) and DLS (Appendix A) show a decrease in particle size as the template in the gel solution is increased. SEM images show a transition in morphology from cubic to hyper-rectangle and then to nanoplates (S3–S8). The larger plane of the plates refers to the (100) plane, which allows stacking and increases the peak intensity at a 2*θ* value of 9.5c. Contrarily, the peaks at 2*θ* values of 14.1, 18.0, 25.8, and 34.4° keep diminishing as the template amount in the synthesis gel was raised and eventually disappeared. The variation in the peak intensities shows a preferred orientation growth of SAPO-34 particles as the template amount is raised. This can be related to a combined effect of two factors introduced by the template. Firstly, the presence of a template promotes the nucleation process, and therefore the number of nuclei in the gel is raised. Secondly, it is widely known that the TEA^+^ cation promotes the formation of the CHA cage [39]. A higher template concentration raises the CHA cage formation and increases the nucleation in the system, which promotes a growth in the [100] direction, forming plate-like SAPO-34 particles. In the case of a smaller amount of template, fewer nuclei are generated and the particles initially grow in the preferred direction. However, the growth transits to the isotropic SAPO-34, which is a more stable phase for SAPO-34. The larger, isotropic SAPO-34 particles are formed through the aggregation and diffusional growth process [40,41].

Figure 3 shows the effect of template concentration on the Si/Al ratio, the P/Al ratio, and Si incorporation in SAPO-34 particles. As the template amount was raised, the P/Al ratio declined while the Si/Al ratio increased. The template, TEAOH, enhances the crystallinity of the product and provides a proton for the reaction generating the Brønsted acid site during Si substitution in the AlPO structure. Therefore, high template concentration facilitated Si incorporation, inducing SAPO-34 particles with a high Si/Al ratio [42].

### 3.2. Effect of P_2_O_5_ Concentration (y) on the Formation of SAPO-34 Particles

Three SAPO-34 particles (S7, S9, and S10) were compared to study the effect of decreasing the P_2_O_5_ concentration (*y*) from 2 to 1. XRD of these samples in Figure 1 (S7) and Figure 4 (S9 and S10) shows a transition from pure crystalline to a mixed phase of amorphous gel and crystalline particles. SEM images in Figure 2 (S7) and Figure 5 (S9 and S10) present their morphology. The SAPO-34 particles exhibit plate-like morphology in these samples with variation in amorphous content varying in the order of S7 < S9 < S10. The decrease in phosphoric acid reduced the crystallinity of the SAPO-34. This is because SAPO-34 preferably crystallizes in slightly acidic or neutral pH conditions which is increased as the P_2_O_5_ content decreases. A similar effect was reported by Liu et al. where a low ratio of P_2_O_5_/Al_2_O_3_ in the gel was found to promote the amorphous phases in the synthesized product after conventional hydrothermal synthesis [43]. The gel with low P_2_O_5_ was used for longer synthesis times (S11 and S12), and pure SAPO-34 particles were eventually obtained after hydrothermal treatment for 24 h (S12). The XRD patterns for the samples synthesized for 7, 12, and 24 h are presented in Figure 4 (S10–S12) and show an amorphous to crystalline phase transformation. It was evident that the reduction of P_2_O_5_ in the synthesis gel increases the crystallization time for the synthesis of high phase purity SAPO-34 particles due to less favorable pH conditions.

To study the effect of the time for phosphoric acid addition to the gel solution, four samples (S13–S16) were synthesized by changing the addition time from 0 to 60 min. The composition of the synthesis gel was maintained at 1Al_2_O_3_:4.0TEAOH:2.0P_2_O_5_:0.4SiO_2_:100H_2_O. XRD patterns for these samples are shown in Figure 4 (S13–S16). All the samples synthesized with the variation in the rate of phosphoric acid addition were crystalline SAPO-34 particles and exhibited prominent XRD characteristic peaks. The peak intensity was showing an identical trend to that of the increase in template and the peak of (100) showed strengthening against the other peaks as the time for phosphoric acid addition was raised, which shows a preferred orientation for crystal growth.

The SEM images of the S13–S16 samples are presented in Figure 5. The particle size was reduced as the time for phosphoric acid addition was increased. The quick addition of phosphoric acid immediately produces three effects: (a) decreases the pH of the gel solution, (b) provides the precursor, and (c) results in an exothermic reaction. The decreased pH and a large amount of precursor provide a favorable environment for nuclei formation, and the heating of the system due to the exothermic reaction facilitating the growth. During the short period of reaction, a small number of nuclei are formed, which uniformly increase in size during the aging of the gel solution and result in large isotropic particles upon synthesis [40]. Contrarily, the slow addition of phosphoric acid restricts the amount of precursor in the gel. As discussed earlier, the SAPO-34 formation is preferred in low pH conditions, and therefore the nucleation is restricted due to high pH conditions with a smaller amount of added phosphoric acid [44]. As the amount of acid is increased in the gel solution, nuclei start to appear in the gel but the growth remains restricted as the thermal effect and amount of precursor in the gel solution remain a barrier. With continued stirring and good distribution of phosphoric acid in the mixture, many nuclei are generated in the mixture, which eventually grow into SAPO-34 particles during hydrothermal synthesis. The transition of morphology is related to the fact that the number of nuclei governs the particle size, and the isotropic morphology is stable for larger particle sizes. The particle sizes for S13–S16 were also evaluated using DLS and are presented in Appendix A and the average particle size for the four samples was 1.10, 0.79, 0.33, and 0.25 µm, respectively. The average particle size for these syntheses is also related to the number of nuclei in the aged gel. Thus, the number of nuclei and nucleation rate were found to be responsible for the transition in size, morphology, and uniformity of the synthesized SAPO-34 particles [45].

FTIR was analyzed to see the effect of the rate of acid addition on the gels and is presented in Figure 6. The prominent deep valley for the hydroxyl ion is present in all the samples (S13–16) between 3200–3400 cm^−1^ as shown in Figure 6a. The magnified region for the characteristic bands for the prepared gels is shown in Figure 6b. The strong peak at 1643 cm^−1^ is from the surface-adsorbed hydroxyl ions. The TO_4_ symmetric and asymmetric stretches of PO_4_ can be noticed at 1071 cm^−1^, 1125 cm^−1^, and 1482 cm^−1^, respectively. The amine-based template in the gel shows a band at 1397 cm^−1^. The band intensity and area for the template and PO_4_ are stronger for the slower addition of phosphoric acids when compared with quick additions. Therefore, the significant amount of PO_4_ in the gel solution for a longer time of acid addition has become part of the gel polymer structure. The binary bands at 470 and 509 cm^−1^ are produced from the SiO_4_ and AlO_4_ tetrahedrons, and the band at 938 cm^−1^ also represents the SiO_4_ tetrahedron. These bands show the opposite behavior to that of PO_4_. As a result, it can be interpreted that the rate of acid addition governs the structural formation of synthesized SAPO-34 particles since it had a significant effect on the gel structure. This was also observed in the chemistry of the synthesized SAPO-34 particles. The Si/Al ratio, P/Al ratio, and Si incorporation calculated from the EDX data are presented in Figure 7. They indicate the trend following the explanations of the FTIR. The slower addition of acid over a longer period initially increased the P/Al ratio of the particles, and then the P/Al ratio stabilized. Both the Si/Al ratio and Si incorporation, however, were found to show opposite behavior. These observations can also be understood by considering that many researchers get SAPO-5 and amorphous phases with thick gel systems [46,47]. The SAPO-5 is a low Si phase, and the amorphous phase forms with segregated Si in the system. With the quick addition of a precursor in the gel solution, heat is generated in the system, which causes a quick reaction and forms regions of high and low Si polymeric formation in a highly viscous medium, resulting in the inhomogeneity of the gel solution. However, slower addition allows a homogeneous gel to form without heating the system, and therefore a uniform composition and structure of SAPO-34 are achieved.

The BET adsorption isotherms for N_2_ adsorption on cubic (S13) and nanoplates (S15) SAPO-34 particles are shown in Figure 8. It can be observed that both the cubic and nanoplate SAPO-34 particles show type I adsorption behavior at lower *p*/*p*_o_ values. The isotherm at higher *p*/*p*_o_ in the S15 sample shows a sharp increase in adsorption due to condensation in the macroposity generated by the stacking of SAPO-34 sheets. The surface area characteristics for the cubic and plate-like particles are presented in Table 3, where it can be clearly seen that the nanoplates exhibit higher BET surface area and total pore volume as compared with the cubic particles.

### 3.3. Effect of SiO_2_ Concentration (z) on the Formation of SAPO-34 Particles

Three SAPO-34 particles with a gel composition of 1Al_2_O_3_:4TEAOH:2P_2_O_5_:*z*SiO_2:_100H_2_O, where *z* = 0.4 to 0.6 (S15, S17, and S8) were analyzed for their structural and chemical properties. All the three compositions used with *z* = 0.4, 0.5, and 0.6 show the plate-like shape of particles in the SEM images (Appendix A). The XRD patterns of these samples show sharp characteristic peaks of SAPO-34 (Appendix A). The silica content doesn’t change the morphology and Si incorporation as shown in Figure 9 whereas the chemical composition of these particles shows significant variations. As the Si content in the gel solution increases, the Si/Al ratio shows a sharp rise, while the P/Al ratio slightly decreases. The EDS results show that Si starts to substitute both P and Al when the SiO_2_ concentration in the gel solution is high.

### 3.4. Density of Acid Sites in SAPO-34 Particles

Representative SAPO-34 samples with diverse morphology and sizes (S5, S7, S8, S13, and S15–S17) were analyzed by NH_3_–TPD to check the density of acid sites. Figure 10 shows the NH_3_ desorption curves, and Table 4 summarizes the density of acid sites and desorption temperatures. The result indicates the presence of a significant amount of both weak and strong acid sites in these samples. The increase in TEAOH and silica concentration (*x*, *z*) in the gel solution shows a rise in the number of acid sites in the synthesized particles, whereas the effect of time for phosphoric acid addition shows a scattered result.

A visual representation of the data is presented in Figure 11 where the variation of acid sites against Si/Al and P/Al ratios is plotted for the samples. It shows that the density of acid sites highly depends on these ratios. This is because more Brønsted acid and base sites are generated in the structure with the homogeneous distribution of Si into the zeolite backbone. The number of acid sites increased with increasing SiO_2_ and TEAOH concentrations. As shown in Figure 3, the template facilitates Si incorporation and raises the Si/Al ratio which improved the density of acid sites. The SiO_2_ concentration was far more significantly affecting the density of acid sites. As previously discussed in Figure 9, the silica concentration doesn’t affect the Si incorporation but made a significant effect on the Si/Al ratio, which is the important factor for governing the density of active sites. The increasing Si/Al ratio for the S15, S17, and S8 samples increased the density of acid sites rapidly. The effect was in reverse for the P/Al ratio, which shows that Si is well substituting Al in the framework of SAPO-34 and therefore more protons are generated in the structure. The rate of phosphoric acid addition shows a shuffle in the result. As shown in Figure 7, the time for the addition of phosphoric acid was found to be inversely related to the incorporation of Si, but a lower density of acid sites was obtained even with a higher Si/Al ratio of product in the case of faster phosphoric addition. The decrease in the acid site can be related to the Si island due to the exothermic irreversible reaction while quick addition of phosphoric acid limits the time for homogenization of the gel [43]. The slower addition of phosphoric acid allows the more uniform precursor distribution and therefore a large number of acid sites are obtained in the final product. A similar observation was made by Lye et al. where reduction of acid sites in SAPO-11 was reported due to the formation of Si-islands [48]. The Si-island is formed with Si surrounded by four silicon atoms which make the enveloped atom electronically stable and unable to support the proton. Larger clusters of Si, therefore, reduce the overall protonic sites and hence the catalytic activity. From Figure 11, it is clear that the Si/Al ratio is the most important characteristic to govern the density of active sites.

A comparison of characteristics of the various reported SAPO-34 particles is presented in Table 5. The cubic particles show a much larger particle size with high acid site densities. The mass transfer resistance of these particles makes them much more prone to coking, which significantly reduces their catalytic lifetime. The hierarchical and plate-like particles allow a much better catalytic lifetime, but their low Si/Al ratio and less amount of catalytic active sites reduce the overall catalytic performance. The SAPO-34 nanoplates synthesized in this study have a high Si/Al ratio compared with previously reported particles of similar morphology. Moreover, they show a much higher acid site density in the structure. It is expected that these SAPO-34 particles will show better catalytic performance with a longer lifetime due to their low mass transfer resistance, higher purity, and higher acid site density.

## 4. Conclusions

Microwave hydrothermal synthesis was used to synthesize SAPO-34 nanoplates with an increased density of acid sites. Various alterations in the gels had significant effects on the morphology and Si distribution of the synthesized particles. First, the higher template concentration reduced the particle size, and increased the Si/Al ratio and Si incorporation simultaneously. Second, higher phosphoric acid addition in the gel achieved smaller particles with higher crystallinity, and adding it slowly enhanced the homogeneity of the gel, resulting in smaller nanoplates and retarding the formation of Si islands. Third, higher silica concentration retained the plate-like morphology with an increase in Si/Al ratio. From the results, it was clearly proven that SAPO-34 nanoplates with a high Si/Al ratio can be synthesized by controlling the composition and homogeneity of the gel solution. The highest density of acid sites for SAPO-34 nanoplates achieved in the present study was 3.48 mmol/g, much greater than those reported in the literature. The high density of acid sites and nanoplate morphology of SAPO-34 is expected to provide higher catalytic activity with lower mass transfer resistance.

## Figures and Tables

**Figure 1 nanomaterials-11-03198-f001:**
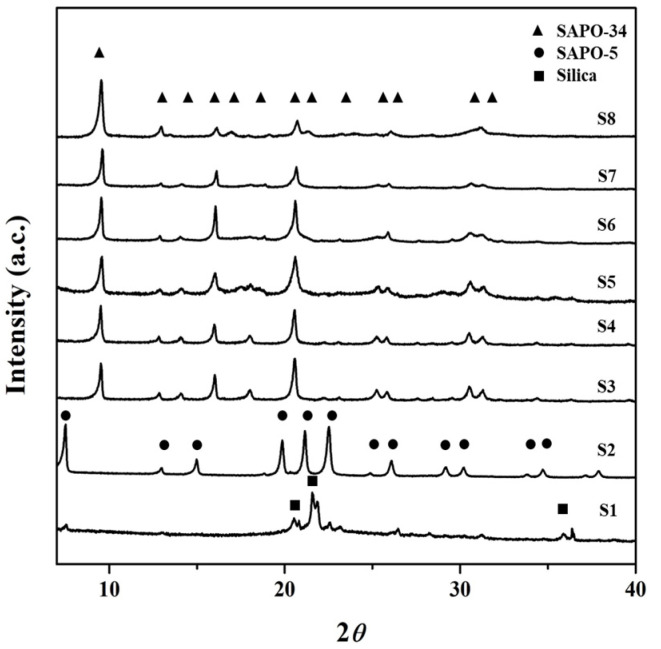
XRD of as-synthesized SAPO-34 particles with variation of template concentration (*x*).

**Figure 2 nanomaterials-11-03198-f002:**
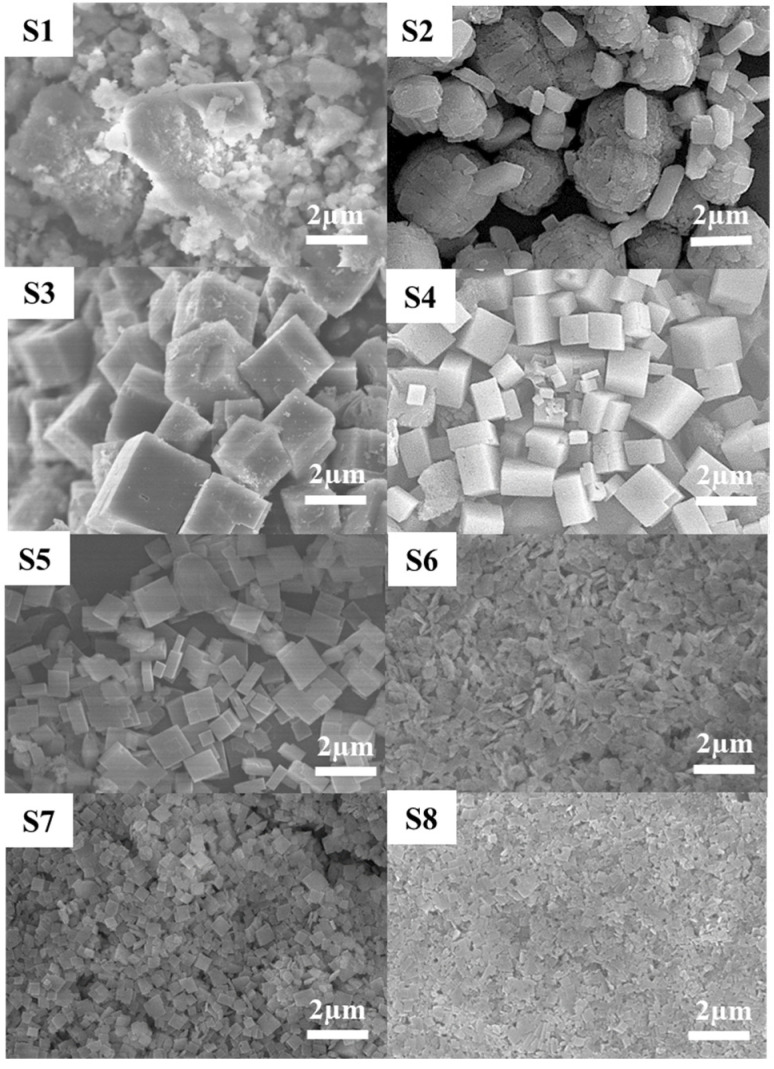
SEM images of the as-synthesized SAPO-34 particles with the variation of template concentration (**S1**–**S8**).

**Figure 3 nanomaterials-11-03198-f003:**
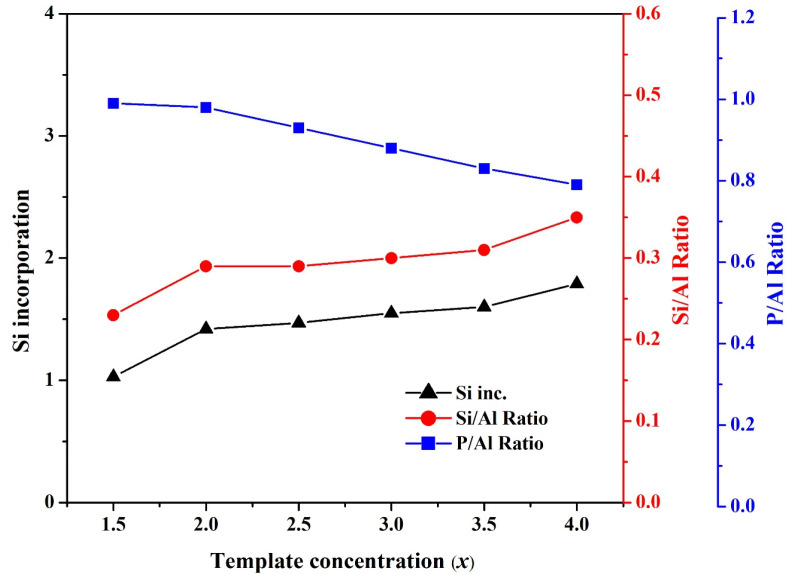
The effect of template concentration (*x*) on the Si/Al ratio, P/Al ratio, and Si incorporation for the synthesized SAPO-34 particles.

**Figure 4 nanomaterials-11-03198-f004:**
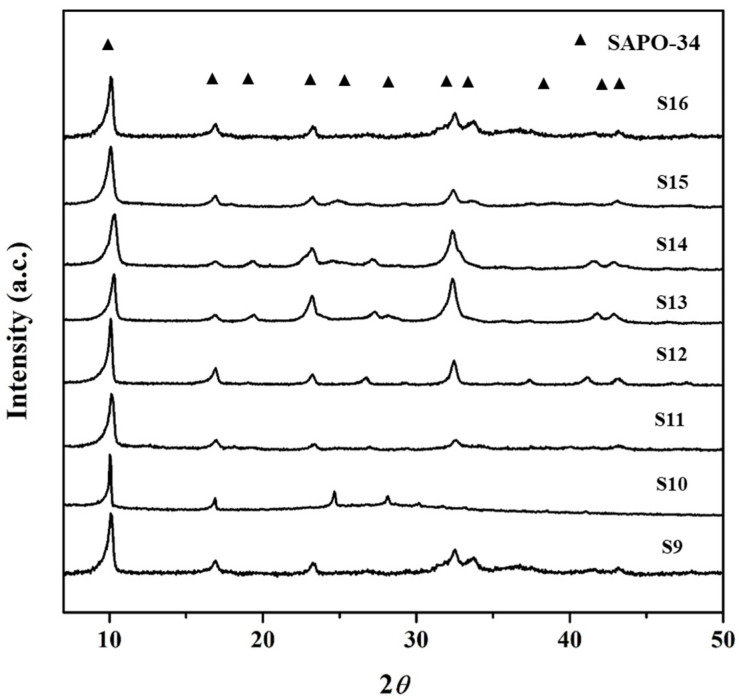
XRD for the as-synthesized SAPO-34 particles (S9–S16) with varied P_2_O_5_ concentration (*y*), synthesis time, and variation in phosphoric acid addition time.

**Figure 5 nanomaterials-11-03198-f005:**
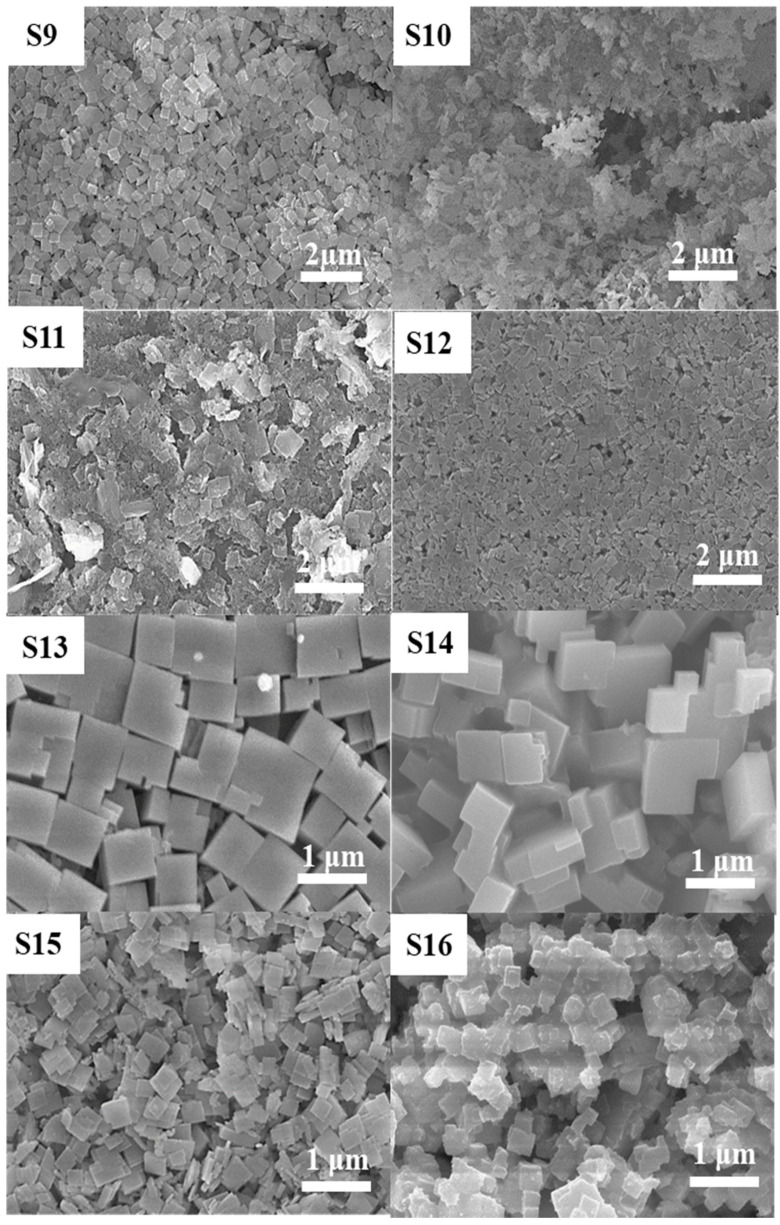
SEM images as-synthesized SAPO-34 particles (**S9**–**S16**) with varied P_2_O_5_ concentration (*y*), synthesis time, and variation in phosphoric acid addition time.

**Figure 6 nanomaterials-11-03198-f006:**
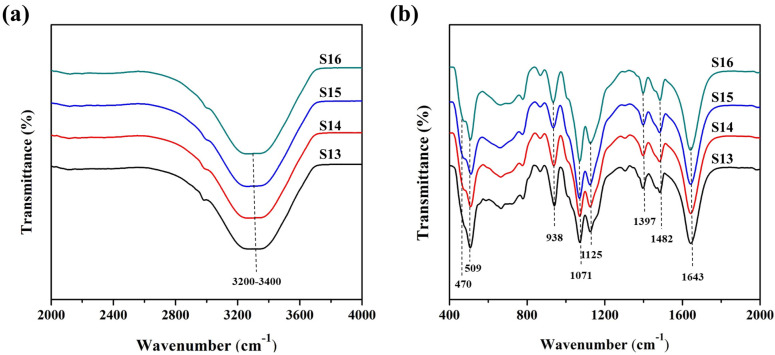
FTIR of the gels prepared with different phosphoric acid addition time (S13–S16): (**a**) 2000–4000 cm^−^^1^, and (**b**) 400–2000 cm^−1^.

**Figure 7 nanomaterials-11-03198-f007:**
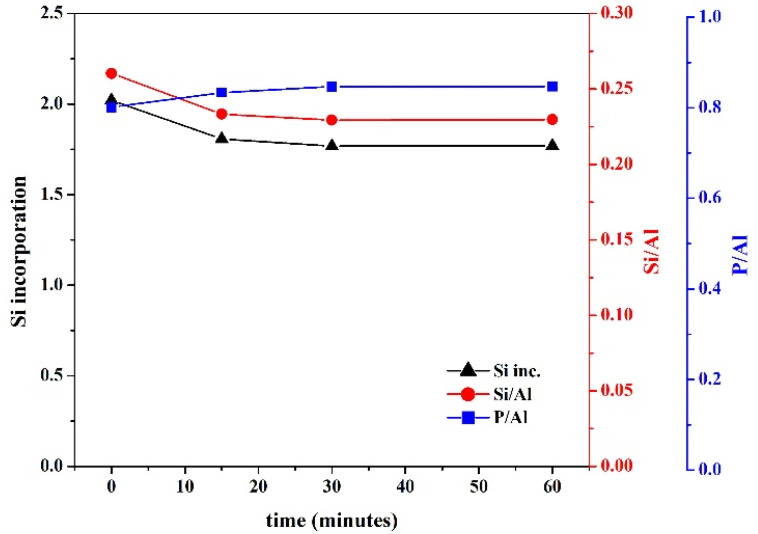
The effect of phosphoric acid addition time on the Si/Al ratio, P/Al ratio, and Si incorporation of synthesized SAPO-34 particles.

**Figure 8 nanomaterials-11-03198-f008:**
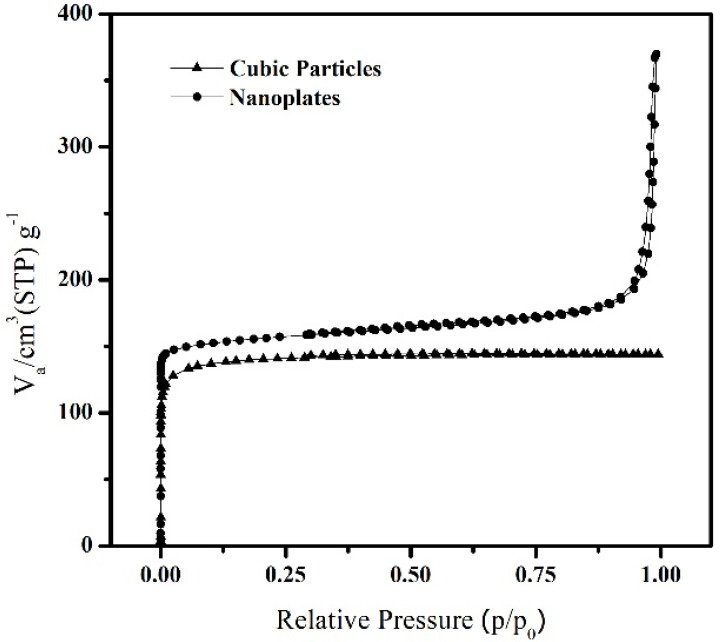
N_2_ adsorption–desorption BET-isotherms of cubic (S13) and nanoplate (S15) type SAPO-34 particles.

**Figure 9 nanomaterials-11-03198-f009:**
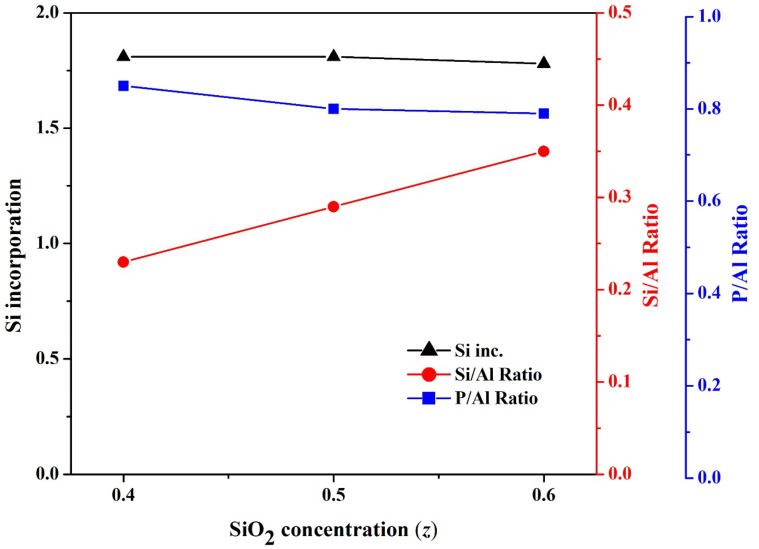
The effect of SiO_2_ concentration (*z*) in the gel solution on the Si/Al ratio, P/Al ratio, and Si incorporation for the synthesized SAPO-34 particles.

**Figure 10 nanomaterials-11-03198-f010:**
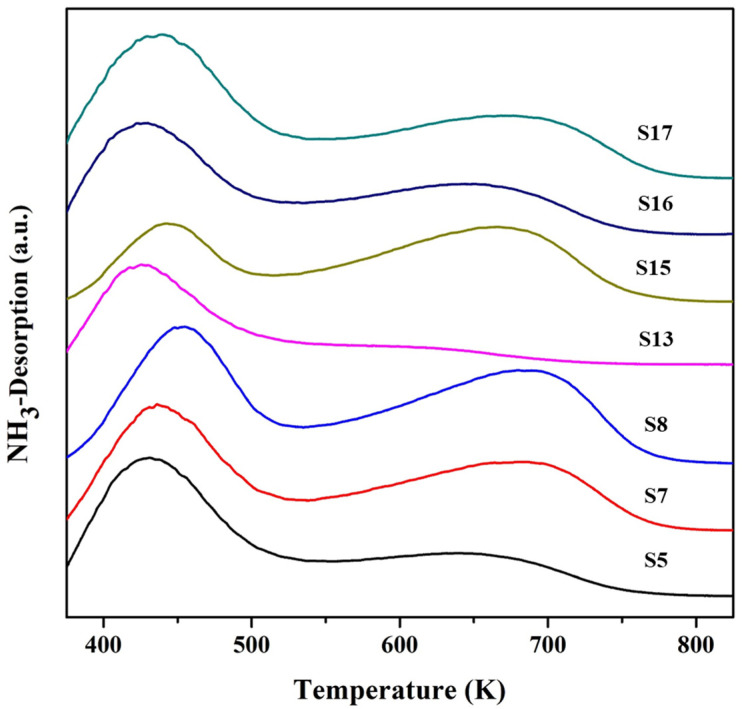
NH_3_-TPD analysis of the synthesized SAPO-34 particles.

**Figure 11 nanomaterials-11-03198-f011:**
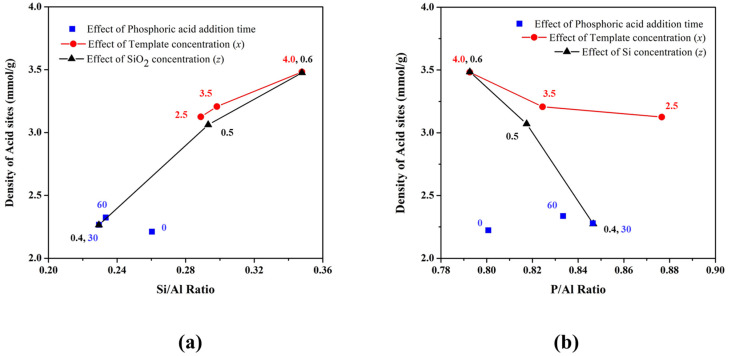
The effect on the density of acid sites with the variation of (**a**) Si/Al ratio and (**b**) P/Al ratio in the SAPO-34 particles synthesized under different variations.

**Table 1 nanomaterials-11-03198-t001:** Detailed synthesis conditions of 17 samples synthesized in the present study.

	Gel Composition (1Al_2_O_3_:*x*TEAOH:*y*P_2_O_5_:*z*SiO_2_:100H_2_O)			Time for Phosphoric Acid Addition	Synthesis	
	*x*	*y*	*z*	min	T (K)	t (h)
S1	0.0	2.0	0.6	30	453	7
S2	1.0	2.0	0.6	30	453	7
S3	1.5	2.0	0.6	30	453	7
S4	2.0	2.0	0.6	30	453	7
S5	2.5	2.0	0.6	30	453	7
S6	3.0	2.0	0.6	30	453	7
S7	3.5	2.0	0.6	30	453	7
S8	4.0	2.0	0.6	30	453	7
S9	3.5	1.5	0.6	30	453	7
S10	3.5	1.0	0.6	30	453	7
S11	3.5	1.0	0.6	30	453	12
S12	3.5	1.0	0.6	30	453	24
S13	4.0	2.0	0.4	0	453	7
S14	4.0	2.0	0.4	15	453	7
S15	4.0	2.0	0.4	30	453	7
S16	4.0	2.0	0.4	60	453	7
S17	4.0	2.0	0.5	30	453	7

**Table 2 nanomaterials-11-03198-t002:** Summary of materials characteristics of the SAPO-34 particle synthesized in the present study.

	Phase	Morphology	Particle Size (µm)
S1	Mixed Phase	Irregular aggregates	-
S2	SAPO-5	Hexagon	3.91
S3	SAPO-34	Cubic	2.24
S4	SAPO-34	Hyper rectangle	1.92
S5	SAPO-34	Nanoplate	0.89
S6	SAPO-34	Nanoplate	0.22
S7	SAPO-34	Nanoplate	0.16
S8	SAPO-34	Nanoplate	0.13
S9	SAPO-34	Nanoplate	0.13
S10	Amorphous + SAPO-34	Amorphous and nanoplate	-
S11	SAPO-34	Nanoplate	0.18
S12	SAPO-34	Nanoplate	0.19
S13	SAPO-34	Cubic	1.10
S14	SAPO-34	Hyper rectangle	0.79
S15	SAPO-34	Nanoplate	0.33
S16	SAPO-34	Nanoparticles	0.25
S17	SAPO-34	Nanoplate	0.24

**Table 3 nanomaterials-11-03198-t003:** Surface area characteristic of the SAPO-34 particles.

	S_BET_ (m^2^ g^−1^)	V_m_ (cm^3^ (STP) g^−1^)	Total Pore Volume (cm^3^ g^−1^)
Cubic	471.59	108.35	0.2227
Nanoplates	601.67	138.24	0.2935

**Table 4 nanomaterials-11-03198-t004:** NH_3_-TPD data for the synthesized SAPO-34 particles.

	Acid Sites				
	A mmol/g	B mmol/g	Total Acid Sites (mmol/g)	T1 (K)	T2 (K)
S5	2.197	0.929	3.126	432.5	669.2
S7	1.666	1.541	3.207	445.8	675.6
S8	2.276	1.206	3.482	452.5	683.1
S13	1.061	1.155	2.216	427.1	657.5
S15	1.611	0.675	2.286	445.8	675.6
S16	1.491	0.838	2.329	430.8	675.9
S17	1.671	1.398	3.069	449.4	672.8

**Table 5 nanomaterials-11-03198-t005:** A comparison of various reported SAPO-34 particles and their characteristics.

Author	Gel Composition	Morphology	Si/Al *	P/Al *	Density of Acid Sites (mmol/g)	Size (µm)	Ref.
Doan et al. 2019	1.0Al_2_O_3_:0.6SiO_2_:1.0P_2_O_5_:3TEAOH:110H_2_O	Cubic	0.32	0.69	2.25	2.5	[29]
Sedighi et al. 2017	1.0Al_2_O_3_:1.0P_2_O_5_:0.4SiO_2_:0.9Mor:1.1TEAOH:60H_2_O	Cubic	0.23	0.84	1.04	0.5	[27]
Zhang et al. 2018	1.0Al_2_O_3_:0.6SiO_2_:1.0P_2_O_5_:1.0Mor:1.0TEA:1.0TEAOH:100H_2_O	Cubic	0.29	0.75	2.06	1–2	[49]
Wang et al. 2019	1.0Al_2_O_3_:1.2P_2_O_5_:2.0TEAOH:0.6SiO_2_:40H_2_O	Cubic	0.35	0.87	3.15	0.7	[50]
Soltani et al. 2019	Al_2_O_3_:0.6SiO_2_:0.5TEAOH:1.5Mor: P_2_O_5_:60H2O	Cubic			3.26	4.8	[51]
Wang et al. 2015	1.0Al_2_O_3_:1.0P_2_O_5_:0.5SiO_2_:2.3DEA:150H_2_O	Aggregates	0.29	0.74	2.90	30	[52]
Sun et al. 2019	7.0TEA:1.0Al_2_O_3_:1.0P_2_O_5_:0.3SiO_2_:15.7H_2_O	Hierarchical	0.21	0.80	0.90	2.5	[22]
Yang et al. 2013	1.0Al_2_O_3_:2.0P_2_O_5_:4.0TEAOH:0.6SiO_2_:140H_2_O	Sheet	0.12	0.92		0.3	[8]
Gong et al. 2020	1.0Al_2_O_3_:1.0P_2_O_5_:1.1SiO_2_:1.9TEAOH:72H_2_O	Low crystalline sheets	0.49	0.84	0.26	0.2–0.6	[53]
Our study	0.5Al_2_O_3_:1.0P_2_O_5_:0.3SiO_2_:2.0TEAOH:50H_2_O	Sheets	0.35	0.79	3.48	0.2	

* The values are calculated using the EDX and compositional analysis data presented in literature.

## Data Availability

Data is available upon request from authors.

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
