# Peer review of "Synthesis of SAPO-34 Nanoplates with High Si/Al Ratio and Improved Acid Site Density"

_nanomaterials, 2021, doi:10.3390/nano11123198_

Round 1

Reviewer 1 Report

The plate-like SAPO-34 particles have been synthesized by many researchers, esp. Ref. 23 and 7, the best synthesis composition (1.0Al2O3 :2.0P2O5 :0.6SiO2 :4.0TEAOH: 100H2O) obtained in this study is very similar to that (1.0Al2O3 :2.0P2O5 :0.6SiO2 :4.0TEAOH: 140H2O) in Ref. 7. The only difference is the water amount. Therefore, the current article just did a detail investigation on the effect of the amount of TEAOH, P2O5 and SiO2 and the addition rate of phosphoric acid. That is, the novel is not very high.

‘ration’ in title should be ‘ratio’.

Table 1: Time for phosphoric acid addition ‘sec’ should be ‘minute’ according to the context.

Table 3: The effective number of the data is too many. The value of Vm should be checked. Usually it is ~0.2 cm3 g-1.

Fig. 6 is not in the correct position (far from the caption).

S15 in Fig. 10 and S14 in Table 4 are not consistent.

Fig. 10: the strong acid is not so much for S5, but it showed pretty high value in Table 4, please show how the peaks were separated in Supporting file.

Table 5: the water amount of the composition 0.5Al2O3 : 1.0P2O5 : 0.3SiO2 : 2.0TEAOH: 70H2O is different from that in Page 4. Which is the correct one?

Table 5: the Si/Al ratio of the sample obtained by the authors is much higher than that of Ref. 7, which has similar synthesis composition. What is the main reason?

Author Response

General comment:  

The plate like SAPO-34 particles have been synthesized by many researchers, esp. Ref. 13 and 7, the best synthesis composition (1Al2O3: 2P2O5: 0.6SiO2: 4.0TEAOH: 140H2O in Ref. 7. The only difference is the water amount. Therefore, the current article just did a detailed investigation on the effect of the amount of TEAOH, P2O5 and SiO2 and the addition rate of phosphoric acid. That is, the novel is not very high

Comment 1:

“Ration” in title should be “ratio”

Author’s Response:

Thank you for the correction, the change has been made as suggested.

Comment 2:

Table 1: Time for phosphoric acid addition “sec” should be “minute” according to text.

Author’s Response:

Thank you for the correction, the change has been made as suggested.

Comment 3:

Table 3. The effective number of data is too many. The value of Vm should be checked. Usually ~0.2 cm3 g-1 is too high.

Author’s Response

Thank you for the comment. Zeolite exhibits much higher pore volume due to the presence of microporous structures. There are reports of SAPO-34 having a pore volume of 0.3 cm3 g- (Li et. al.) with other zeolite including ZSM-5 and MFI having a pore volume of as high as 0.9 cm3 g-.

Our data is directly taken from the analysis of BET results. The table shows the characteristics of the prepared samples as analyzed.

Reference

  1. Liping, C. Xiaojing, L. Junfen, and W. Jianguo, “Synthesis of sapo-34/zsm-5 composite and its catalytic performance in the conversion of methanol to hydrocarbons,” J. Braz. Chem. Soc., vol. 26, no. 2, pp. 290–296, 2015, doi: 10.5935/0103-5053.20140279

Comment 4:

Fig. 6 is not in the correct position

Author’s Response

Thank you for the correction, the change has been made as suggested.

Comment 5:

S15 in Fig10 and S14 are not consistent

Fig. 10: the strong acid is not so much for S5 but it shows pretty high value in Table 4, please show how the peaks were separated in Supporting file

Author’s Response

Thank you for highlighting the issue. The analysis of the sample was rechecked. The results are added in the supporting file. The results are consistent with the previous explanations and didn’t require any changes.

Comment 6:

Table 5: the water amount of the composition 0.5Al2O3: 1P2O5: 0.3SiO2: 2.0TEAOH: 70H2O is different from that in Page 4. Which is correct?

Author’s Response

Thank you for the correction, the correct value is 100 H2O at page 4. The value is corrected in the Table 5 accordingly.

Comment 7:

The Si/Al ratio of the sample obtained by the author is much higher than of Ref. 7 which has similar synthesis composition. What is the main reason?

Author’s Response

Thank you for the response. According to our results in FTIR and the effect of composition variation, we think that the gel characteristics significantly influences the Si/Al ratio of the final product. One of the reason that can differentiate the two results is the homogenization process during gel preparation which can alter the Si/Al ratio and Si incorporation into the structure of SAPO-34.

Reviewer 2 Report

The article is very interesting; it is about a SAPO-34 nanoplates synthesis with a high Si/Al ratio and improving the acid site density, which is very important to catalysis application.

I have some suggestions and questions, as follow:

1) The authors comment that numerous researchers tried to synthesize the SAPO-34 nanoplates (page 2, lines 54 and 55). What is the difference of the article with these researches works? Is it the high Si/Al ratio with the limited formation of Si islands? (page 3, line 102).

2) On page 2, lines 71, 72, and 73, the authors present the mechanism of Si substitution. The elements that are in [ xxx ] are in the framework? If yes, maybe it is interesting to Clare up that.

3) Do not cut tables and figures. See Table 1, Figure 4, and Table 4. Figure 6 is displaced.

4) Comparing Table 3 with  Figure 2 and Figure 5; it is not clear the morphology of the particles. For example, S3 is cubic, ok. S4 is hyper-rectangle but also some cubic too. S5 to S8 are so different and are nanoplates? I am not seeing nanoplates.... Maybe TEM analysis will be better than SEM to see nanoplates?

Also on page 8  in figure 5, again have "plate-like morphology" (line 233) It is not clear that is plate morphology.

5) In FIgure 3, the axle is the P/Al ratio and the legend is the Al/P ratio. What is correct? This also occurs in Figure 9. Confirm also Figure 7.

6) Page 10, lines 321 and 322. S13 is cubic and S16 nanoplates? It is not clear.

7) Figure 8, comparing S13 and S16. In the S16 the particle sizes are more little than S13, so this sample will have more contribution on the external superficial area. But it is not indicative of plate-like morfology.

8) The isotherms are very well done, congratulations, it is possible to see B point. Both isotherms have type I in low p/po. They virtually type I. The S16 isotherm, increases in high p/po that can be external area contribution or intraparticle porous (macroporosity), or condensation on the glass support. IT IS NOT TYPE IV.

9) Also in Figure S3, it is not possible to see plate-like morphology.

10) The density of acid sites is given by mmol/g. To call density is correct to use mmol or g/volume?  Because density gives some distribution notion.

11) It is possible to put in Table 5 the SEM or TEM from each study giving the morphology type?

12) The  goal of the article was:

  • synthesize nanoplates SAPO-34,  for me it is not clear that authors achieve that.
  • High Si/Al ratio, that is ok
  • limited formation of Si islands, where or how it is measured?
  • the SAPO-34 nanoplates are expected to serve as a high performance catalyst due to the low transfer resistance and high density of active sites. (last line from the abstract); ok, but the authors imagine how to measure transfer resistance or catalytic properties?

13) The references are ok, actual, and adequate, however, I suggest putting all authors' names, do not use et al in the references list.

Author Response

General Comments:

The article is very interest; it is about a SAPO-34 nanoplates synthesis with a high Si/Al ratio and improving the acid site density, which is very important to catalyst application.

I have some suggestions and questions, as follows;

Comment 1:

The author comment that numerous researchers tried to synthesized the SAPO-34 nanoplates (page 2, line 54 and 55). What is the difference of the article with these researches work? Is it the high Si/Al ratio with the limited formation of Si islands? (page 3, line 102).

Author’s Response

Thank you for the response. Yes, the author thinks that the primary finding is the increased Si/Al ratio and acid site density in the synthesized SAPO-34 particles which are primary factor to govern catalytic performance. The improvement in the acid site density is not only influenced by the Si/Al ratio but the distribution of Si in the structure. In the current study, the controlled gel formation has improved the homogeneity of gel which has resulted in substantial improvement in the characteristics of synthesized SAPO-34 particles

Comment 2:

On page 2, lines 71, 72, 73, the authors present the mechanism of Si substitution. The elements that are in [xxx] in the framework? If yes, maybe it is interesting to clare up in the

Author’s Response

Thank you for the suggestion. Yes, the [] actually highlight the backbone of zeolite structure where the substitution takes place. The change has been made as suggested.

Comment 3:

Do not cut tables and figures. See Table 1, Figure 4 and Table 4. Figure 6 is displaced.

Author’s Response

Thank you for the response. The manuscript is revised as suggested.

Comment 4:

Comparing Table 3 with Figure 2 and Figure 5; it is not clear the morphology of the particles. For example, S3 is cubic, okay, S4 is hyper-rectangle but also some cubic too. S4 to S8 are so different and are nanoplates…. I am not seeing nanoplates… Maybe TEM analysis will be better than SEM to see nanoplates?

Also on page 8 in figure 5, again have “plate-like morphology (line 233) It is not clear that is plate morphology.

Author’s Response

Thank you for the response. These particles exhibit a high aspect ratio, which can be observed from the particles at a lateral angle in the SEM images.

Comment 5:

In Figure 3 the axle is the P/Al ratio and the legend is Al/P ratio. What is correct. This also occurs in Figure 9. Confirm also Figure 7.

Author’s Response

Thank you for the comment. The captions in the Figure 3, 7 and 9 are checked and corrected to P/Al, which is the correct notation.

Comment 6:

Page 10, line 321 and 322. S13 is cubic and S16 nanoplates? It is not clear

Author’s Response

Thank you for the comment. The sample S15 was tested for the BET. It has been corrected in the manuscript.

Comment 7:

Figure 8, comparing S13 and S16. In the S16 the particle sizes are more little than S13. So this sample will have more contribution on the external surface area. But it is not indicative of plate-like morphology.

Author’s Response

Thank you for the comment. We have referred the plate-like morphology to explain the results obtained from BET. The discussion is improved in the explanation of the Fig. 8.

Comment 8:

The isotherms are very well done, congratulations, it is possible to see B point. Both isotherms have type I in low p/po. They virtually type 1. The S16 isotherm increases in high p/po that can be external area contribution of intra particle porous (macro porosity), or condensation on the glass support. IT IS NOT TYPE IV.

Author’s Response

Thank you for appreciation and guidance. The BET explanation is improved and corrected in for Fig. 8 as suggested.

Comment 9:

Also in Figure S3, it is not possible to see the plate like morphology.

Author’s Response

Thank you for the response. Similar to Figure 2, the Figure S3 also has particles in lateral angles can be seen in the SEM to identify the plate-like structure.

Comment 10:

The density of acid site is given by mmol/g, to call density is correct to use mmol of g/volume. Because density gives some distribution notion.

Author’s Response

Thank you for the response. The term “density of acid sites” is usually referred to as the amount of acid sites per gram of a catalyst in mmol/g. Therefore, we mentioned the values using the units commonly used in literature.

Comment 11:

It is possible to put table 5 the SEM, or TEM from each study giving the morphology type?

Author’s Response

Thank you for the suggestion. We have evaluated the table 5 with the SEM image but it is very complex to incorporate all the data and SEM images in that table. So we are keeping the current outline with only mentioning the particle sizes of the reported references.

Comment 12:

The goal of the article was:

High Si/Al ratio, that is okay

Limited formation of Si islands, where or how it is measured?

The SAPO-34 nanoplates are expected to serve as a high performance catalyst due to low transfer resistance and high density of acid sites. (last line form the abstract); ok, but the author imaging how to measure the resistance or catalytic properties?

Author’s Response

Thank you for the suggestion. Theoretically, the properties of synthesized SAPO-34 particles show much positive characteristics for an efficient catalyst as it meets the required factor of high acid site density and lower mass transfer resistance as mentioned by earlier researcher. We are working on the study about the catalytic demonstration of these particles and plan to present the results in future.

Comment 13:

The references are all okay, actual, and adequate, however, I suggest putting all authors’ names, do not use et al. in the reference list.

Author’s Response

Thank you for the response. We had generated the reference directly through “Mendeley”.  The references are modified as suggested.

Reviewer 3 Report

In this work, high purity of SAPO-34 nanoplates with increased density of acid site were synthesized using microwave hydrothermal synthesis. The effect of various parameters such as concentrations of structure directing agent, phosphoric acid, and silicon in the gel solution on the phase, shape, and composition of the synthesized samples was investigated. The results showed that the highest density of acid sites for SAPO-34 nanoplates could be obtained by controlling the composition and homogeneity of gel solution. These might provide some helpful references for preparing the novel SAPO-34 materials. The specific questions or advices are as follows.

1.In Fig.10, the S13 sample exhibited a little amount of strong acid sites. However, the substantial amounts of acidic sites were still listed in Table 4. How to explain this discrepancy?

2.The layout of Fig.6 was informal. It is suggested to rearrange its position.

3.There are many small paragraphs in the manuscript. It is suggested to integrate them.

4.In line 228, the authors said“ 1 (S7) and Fig. 4 (S9-227 S10) shows a transition from pure crystalline to mixed phase”. What is the mixed phase? Please further clarify it.

5.There are lots spelling and grammar errors such as “ration”in title, “syntheses”in line 113 and 123, “and hence is the catalytic activity” in line 63 and “is directly related to the catalytic activity was studied”. Please carefully check the similar mistakes and amend them. 

Author Response

General comments:

In this work, high purity of SAPO-34 nanoplates with increased density of acid sites were synthesized using microwave hydrothermal synthesis. The effect of various parameters such as concentration of structure directing agent, phosphoric acid and silicon in the gel solution on the phase, shape and composition of the synthesized samples was investigated. The results showed that the highest density of acid sites for SAPO-34 nanoplates could be obtained by controlling the composition and homogeneity of gel solution. These might provide some helpful references for preparing the SAPO-34 materials. The specific questions or advices are as follows.

Comment 1:

In Fig. 10, the S13 sample exhibits a little amount of strong acid sites however the substantial amount of acid sites was still listed in Table 4. How to explain this discrepancy?

Author’s Response

Thank you for the response. The result for the sample was reevaluated and screenshot is inserted in supporting data. The results are consistent with the previous explanations and didn’t require any changes.

Comment 2:

The layout of Fig. 6 was informal. It is suggested to rearrange its position.

Author’s Response

Thank you for the response. The figure has been repositioned according to suggestion.

Comment 3:

There are many small paragraphs in the manuscripts. It is suggested to integrate them.

Author’s Response

Thank you for the response. The manuscript is revised as suggested.

Comment 4:

In line 228, the author said” 1 (S7) and Fig. (S9-227S10) shows a transition from pure crystalline to mixed phase”. What is the mixed phase please further clarify it?

Author’s Response

Thank you for the comment. The mixed phase refers to presence of both crystalline and amorphous phase. The sentence has been edited to make it clear to the readers.

Comment 5:

There are a lot of spelling and grammar mistakes errors such as ‘ratio’ in the title, ‘Syntheses in line 113 amd 123, “and hence in the catalytic activity” in line 63 and “is directly related to catalytic activity was studied.” Please carefully check similar mistakes and amend them.

Author’s Response

Thank you for the response. The entire manuscript is reviewed and rechecked for errors and is improved as suggested.

Reviewer 4 Report

In my opinion, the paper “Synthesis of SAPO-34 Nanoplates with High Si/Al ration and 2 Improved Acid Site Density” by Syed Fakhar Alam et al. does not present new information to be published in a ranked journal as Nanomaterials. I suggest the authors to re-submit to other lower ranked journal.

Minor remarks:

  • Figure 6 it is out of place
  • The nitrogen adsorption isotherm (Figure 8) for plates can not be classified simply as “type IV”. Please check IUPAC information.

Author Response

General Comments:

In my opinion, the paper “Synthesis of SAPO-34 Nanoplates with High Si/Al ration and Improved Acid Site Density” by Syed Fakhar Alam et al. does not present new information to be published in a ranked journal such as Nanomaterials. I suggest the author to resubmit to other lower ranked journal.

Minor Remarks:

Figure 6 it is out of place.

The nitrogen adsorption isotherm (figure 8) for plates cannot be classified simply as “type IV”. Please check IUPAC information

Author’s Response:

Thank you for the response. We have improved the manuscript as the reviewer suggested.

Round 2

Reviewer 1 Report

The authors have carefully revised the manuscript, therefore, it can be published now.

Reviewer 2 Report

Thanks for having responded to all my queries; I am satisfied. The manuscript can be accepted in this actual form.

Reviewer 4 Report

I suggested the authors to re-submit to other lower ranked journal but, if the Editor accepts its OK by my side.